# T-Cell Defects Associated to Lack of Spike-Specific Antibodies after BNT162b2 Full Immunization Followed by a Booster Dose in Patients with Common Variable Immune Deficiencies

**DOI:** 10.3390/cells11121918

**Published:** 2022-06-14

**Authors:** Federica Pulvirenti, Stefano Di Cecca, Matilde Sinibaldi, Eva Piano Mortari, Sara Terreri, Christian Albano, Marika Guercio, Eleonora Sculco, Cinzia Milito, Simona Ferrari, Franco Locatelli, Concetta Quintarelli, Rita Carsetti, Isabella Quinti

**Affiliations:** 1Reference Centre for Primary Immune Deficiencies, Azienda Ospedaliera Universitaria Policlinico Umberto I, 00185 Rome, Italy; federica.pulvirenti.md@gmail.com; 2Department Onco-Haematology, and Cell and Gene Therapy, Bambino Gesù Children Hospital, IRCCS, 00116 Rome, Italy; stefano.dicecca@opbg.net (S.D.C.); matilde.sinibaldi@opbg.net (M.S.); marika.guercio@opbg.net (M.G.); franco.locatelli@uniroma1.it (F.L.); concetta.quintarelli@opbg.net (C.Q.); 3B Cell Unit, Immunology Research Area, Bambino Gesù Children’s Hospital, IRCCS, Viale di San Paolo, 00146 Rome, Italy; eva.pianomortari@gmail.com (E.P.M.); sara.terreri87@gmail.com (S.T.); christian.albano@opbg.net (C.A.); rita.carsetti@opbg.net (R.C.); 4Department of Molecular Medicine, Sapienza University of Rome, 00185 Rome, Italy; elenorasculco31@gmail.com (E.S.); cinzia.milito@uniroma1.it (C.M.); 5Medical Genetics Unit, IRCCS Azienda Ospedaliero-Universitaria di Bologna, 40138 Bologna, Italy; simo.ferrari@unibo.it

**Keywords:** Common Variable Immune Deficiencies, T-cells, SARS-CoV-2, COVID-19, BNT162b2, vaccine, booster dose, memory B cells, spike protein, antibody response

## Abstract

Following the third booster dose of the mRNA vaccine, Common Variable Immune Deficiencies (CVID) patients may not produce specific antibodies against the virus spike protein. The T-cell abnormalities associated with the absence of antibodies are still a matter of investigation. Spike-specific IgG and IgA, peripheral T cell subsets, CD40L and cytokine expression, and Spike-specific specific T-cells responses were evaluated in 47 CVID and 26 healthy donors after three doses of BNT162b2 vaccine. Testing was performed two weeks after the third vaccine dose. Thirty-six percent of the patients did not produce anti-SARS-CoV-2 IgG or IgA antibodies. Non responder patients had lower peripheral blood lymphocyte counts, circulating naïve and central memory T-cells, low CD40L expression on the CD4+CD45+RO+ and CD8+CD45+RO+ T-cells, high frequencies of TNFα and IFNγ expressing CD8+ T-cells, and defective release of IFNγ and TNFα following stimulation with Spike peptides. Non responders had a more complex disease phenotype, with higher frequencies of structural lung damage and autoimmunity, especially autoimmune cytopenia. Thirty-five percent of them developed a SARS-CoV-2 infection after immunization in comparison to twenty percent of CVID who responded to immunization with antibodies production. CVID-associated T cell abnormalities contributed to the absence of SARS-CoV-2 specific antibodies after full immunization.

## 1. Introduction

Antibodies and memory B cells defects account for the inability to mount adequate antibody responses to infections and vaccinations and for the susceptibility to vaccine preventable infections in Common Variable Immunodeficiency (CVID) patients. The Committee of Experts on Primary Immunodeficiency of the International Union of Immunological Societies included specific antibody responses to vaccinations as a diagnostic tool, and as a means of prevention [1].

Since the start of the severe acute respiratory syndrome coronavirus 2 (SARS-CoV-2) pandemic, CVID patients have been listed as a fragile population [2,3] and have been prioritized for SARS-CoV-2 vaccination. Differently from other microorganisms, SARS-CoV-2, not encountered before by humans, offers the possibility to study immune responses to a novel antigen, and to generate information useful for defining the mechanisms underlying most infectious and inflammatory diseases.

We already identified a peculiar and atypical signature of B cell-mediated immune response to SARS-CoV-2 immunization [4,5]. However, besides the impaired production of specific antibodies and memory B cells, abnormalities of T cell responses may also play an important role in vaccine failure. CD4+ T-cell lymphopenia is associated with an increased risk of complications in CVID [6], as well as cytokine alterations, lack of IL-2, increased TNFα, IFNγ abnormalities with an IFN mRNA signature [7,8], mainly described in CVID patients with an inflammatory phenotype.

A detailed analysis of CVID-associated T-cell defects contributing to the impaired antibody response to SARS-CoV-2 immunization in CVID is lacking. Recently, a low peripheral T-cell count before vaccination, and a defect of T-helper follicular T-cells was found to be associated with lower anti-S1 IgG levels after immunization [9]. Our aim was to identify possible T-cell biomarkers associated with the failure to produce SARS-CoV-2 specific antibodies after full immunization plus booster dose of mRNA vaccine. To assess this need, we studied a group of CVID patients never infected by SARS-CoV-2 who completed the full immunization with the BNT162b2 vaccine followed by a booster dose six months apart. In this cohort, we investigated whether vaccine antibody non-responsiveness was associated with abnormalities of peripheral T-cells lymphocytes distribution and function.

## 2. Materials and Methods

### 2.1. Study Design and Patients

An observational study was carried out in 47 CVID patients naïve to SARS-CoV-2 infection who agreed to undergo immunization with the BNT162b2 vaccine. Participants were diagnosed as having CVID according to the ESID criteria (www.ESID.com accessed on 1 June 2022) and were regularly followed by the Care Center for adults with PID in Rome, Italy. Twenty-six age matched healthy subjects were studied as controls. Eligible patients were informed on the study, including its safety profile and supply procedures. The BNT162b2 vaccine was administered as prescribed in two doses 21 days apart, followed by a booster dose after four months. Blood samples were obtained for serological assessment at baseline (BL), fifteen days after the second dose (2D) and fifteen days after the third dose (3D). Blood samples for the study of cellular immunity were obtained fifteen days after the third dose (3D).

At BL, a set of variables was recorded for each patient including gender, date of birth, IgA and IgM serum levels at study time, and clinical PID-related disorders, including autoimmunity, bronchiectasis and chronic lung disease (granulomatous lung interstitial disease [GLILD], obstructive chronic lung disease [COPD], interstitial lung disease). During the study, participants were allowed to continue their treatments, including immunoglobulin substitution as the standard therapy for underlying antibody deficiency. The study was approved by the Ethical Committee of the Sapienza University of Rome (Prot. 0521/2020, 13 July 2020) and performed in accordance with the Good Clinical Practice guidelines, the International Conference on Harmonization guidelines, and the most recent version of the Declaration of Helsinki.

### 2.2. Anti-Spike IgG and IgA antibodies

A semi-quantitative in vitro determination of human IgG antibodies against the SARS-CoV-2 (IgG S1) and of human IgA antibodies against the SARS-CoV-2 (IgA S1) was performed on serum samples by using the Anti-SARS-CoV-2 Spike ELISA (EUROIMMUN), according to the manufacturer’s instructions. Values were then normalized for comparison with a calibrator. Results were evaluated by calculating the ratio between the extinction of samples and the extinction of the calibrator. Results are reported as the ratio between OD sample and OD calibrator. The ratio interpretation was as follows: <0.8 = negative, ≥0.8 to <1.1 = borderline, ≥1.1 = positive.

### 2.3. Cell Isolation and Cryopreservation, T and B Cell Phenotypes

Peripheral blood mononuclear cells (PBMCs) were isolated by Ficoll Paque™ Plus 206 (Amersham Pharmacia Biotech, Amersham, UK) density-gradient centrifugation and immediately frozen and stored in liquid nitrogen until use. The freezing medium contained 90% Fetal Bovine Serum (FBS) and 10% DMSO. Cells were stained with the appropriate combination of fluorochrome-conjugated antibodies to identify T-cell and B-cell subsets according to standard techniques. B-cell populations were identified as CD19+. Memory B cells (MBC) were gated as CD19+CD24+CD27+, and IgM and switched memory B cells were discriminated by the expression of IgM. T cell subsets were identified based on the expression of CD3, CD4, CD8, CD45RO and CCR7 markers by flow cytometry. An analysis of T cell subsets was performed in all 47 CVID patients and in 9 age-matched HD.

Immune cells were identified as follows: naïve CD4+ T cells (CD4+CD45RO−CCR7+), central memory CD4+ (CD4+CD45RO+CCR7−) T cells, naïve CD8+ T cells (CD8+CD45RO-) and central memory CD8+ T cells (CD8+CD45RO+CCR7−). The CD4+ naïve/memory ratio was calculated as a ratio of CD4+CD45RO−CCR7+ and CD4+CD45RO+CCR7−; the CD8+ naïve/memory ratio was calculated as a ratio of CD8+CD45RO−CCR7+ and CD8+CD45RO+CCR7−. Cells were acquired on a BD FACSymphony A5™. Data were analyzed with FlowJo ver. 10.

### 2.4. Functional Studies

PBMCs isolated from CVID patients and HDs were cultured in 96-well cell plates at 1 × 10^6^ cells/well concentration in RPMI 1640 culture medium containing 5% AB serum. PBMCs were cultured at 37 °C, 5% CO_2_, with pools of SARS-CoV-2 viral peptides (pepMix SARS-CoV-2, Spike Glycoprotein, PM-WCPV-S-1, JPT) at the final concentration of 1µg/mL, or with CytoStim^®^ (SARS-CoV-2 Prot_S PBMC Kit, Miltenyi Biotech, Bergisch Gladbach, Germany) as a positive control or with PBS (Capricon Scientific, Ebsdorfergrund, Germany) as a negative control, for a total of 6 h. After 2 h of incubation, Brefeldin A (SARS-CoV-2 Prot_S PBMC Kit, Miltenyi, Biotech) was added to the cells to inhibit transport of proteins to the cellular membrane. At the end of the stimulation, cells were processed as indicated by SARS-CoV-2 Prot_S PBMC Kit protocol. Briefly, cells were fixed, permeabilized and stained with CD3, CD4, CD8, IFN-γ, TNF-α and CD154 (CD40L) (SARS-CoV-2 Prot_S PBMC Kit protocol, Miltenyi Biotech, Bergisch Gladbach, Germany), with the addition of CD45RO and CD197 (CCR7) antibodies (BD Biosciences, San Diego, CA, USA). 

### 2.5. Statistical Analysis

Data were summarized with descriptive statistics (median and IQR for continuous values, and number (*n*) and frequency for discontinuous variables). Data obtained were compared in CVID and HD. Anti S1 IgG and IgA were also compared between the different study times. Comparison of clinical, immunological and demographic characteristics of CVID participants grouped on the basis of anti S1 IgG- and IgA-antibody production following the third vaccination dose were also made. Univariate analysis assessed the impact of variables of interest. Values were compared by the non-parametric Kruskal–Wallis test and, if not significant, the Wilcoxon matched pair signed-rank test or the two-tailed Mann–Whitney U-test were used. Comparison of clinical characteristics among groups was determined using a two-tailed Chi-square test. Comparison for timing to COVID-19 infection after the third dose of vaccine administration in the different groups was assessed by using Kaplan–Meier product-limit estimates and based on a log-rank and Gehan–Breslow–Wilcoxon test.

Differences were deemed significant when *p*  <  0.05. Statistical Package for Social Sciences version 15 (SPSS Inc., 233 South Wacker Drive, 11th Floor, Chicago, IL, USA) was used for the analysis.

## 3. Results

*Patients*. Forty-seven adult CVID patients diagnosed according to ESID definitions (https://esid.org/Working-Parties/Registry-Working-Party/Diagnosis-criteria accessed on 1 June 2022) were enrolled in the study (median age in years (IQR): 52.3 (47.7–57.17); 57% females). All were naive for SARS-CoV-2 infection, and all were under subcutaneous, intravenous or facilitated subcutaneous Ig substitution therapy. A control group of 26 healthy donors (HD) (median age in years (IQR): 45 (34–52); 65% females) were also included. Pathogenic monogenetic variants known to be associated with CVID were analyzed by targeted gene sequence analysis (Illumina technology performed on a MiSeq NGS). We identified a heterozygous pathogenic variant of PI3Ks gene in a male patient with recurrent infections, COPD, lymphomas and GLILD (c.1573G > A; *p*.Glu525Lys) and a heterozygous pathogenic variant of the TNFRSF13B gene in one patient with non-malignant lymph node enlargement and recurrent infections. Moreover, five patients were found to carry variations of unknown significance (VUS): one patient carried a heterozygous VUS in the TNFRSF13C (c.62C > G; *p*.Pro21Arg) and TNFRSF13B genes (c.515G > A; *p*.Cus172Tyr) plus a heterozygous pathogenic variant of the LRBA gene (c.6433C > T; *p*.Arg2145Cys); one carried a heterozygous VUS in the PLCG2 gene (c.3368A > C; *p*.Glu1123Ala); one carried a heterozygous VUS in the DCLRE1C (c.1561C > A; *p*.Leu521Ile) and the IGLL1 genes (c.301G > A; *p*.Gly110Arg); one patient carried a heterozygous VUS in the STX11 gene (c.616G > A; *p*.Glu206Lys) and a compound heterozygous mutation in the TNFRSF13C gene (c.62C > G; *p*.Pro21Arg/c.475C > T; *p*.His159Tyr); and one patient carried a heterozygous VUS in the FAS gene (c.580G > A; *p*.Glu194Lys).

*SARS-CoV-2 antibodies.* After full immunization with two doses of the mRNA BNT162b2 vaccine, all HD developed anti-S1 IgG and anti-S1 IgA, with levels rising significantly after the booster dose. In CVID, 20% of the patients had measurable anti-S1 IgG after immunization with two doses and 64% after the booster dose, with levels rising after the booster dose (geometric mean [95%CI]: BL: 0 OD ratio (0–0); 2D: 0.38 OD ratio (0.16–0.76); 3D: 2.65 OD ratio (1.42–4.94)) (Figure 1A). The anti-S1 IgA response was less frequently detectable (Figure 1B): only 30% of patients developed anti-S1 IgA after full vaccination. After the booster dose, the proportions of anti-S1 IgA responders and anti-S1 IgA serum levels remained stable (geometric mean [95%CI]: BL: 0.06 OD ratio (0.04–0.15); 2D: 0.22 OD ratio (0.09–0.56); 3D: 0.37 OD ratio (0.21–0.68)). In contrast, anti-S1 IgA was detectable in all HD after full vaccination, reaching higher levels after the booster dose (Figure 1B). Based on antibody responses after the booster dose, we identified three CVID groups: 17 patients who did not develop either anti-S1 IgG nor anti-S1 IgA were defined as non-responders (NR); 18 patients responded with anti-S1 IgG only (S1 IgG-R), and 12 patients responded with anti-S1 IgG and anti-S1 IgA (S1 IgG/IgA-R) (Figure 1C).

As shown in Table 1, in comparison to S1 IgG-R and S1 IgG/IgA-R, CVID NR had a significantly lower count of peripheral B-cells, memory B cells (MBCs), IgM MBCs, and switched MBCs. In contrast, there was no difference between CVID NR, CVID S1 IgG-R and CVID S1 IgG/IgA-R with respect to age at diagnosis, serum IgA and IgM levels, and distribution between the sexes.

Moreover, patients in the NR group in comparison to IgG/IgA antibody responders showed higher frequencies of autoimmune manifestations, including cytopenias (*p* = 0.049) and bronchiectasis (*p* = 0.011) (Figure 2A), revealing a more complex clinical phenotype. During the follow up, 15 patients were infected by SARS-CoV-2 after being immunized with the three doses of BNT162b2 vaccine. Infected patients had lower S1 IgG serum levels in comparison to those who remained free from infection during the study time (0.13 OD ratio (IQR 0.08–0.31) vs. 2.74 OD ratio (IQR 0.5–27.41), *p* < 0.0001), with 8/15 of infected participants belonging to the NR group and four and three patients belonging to the IgG-R and IgG/IgA-R groups, respectively. SARS-CoV-2 infection course was asymptomatic/mild in all infected patients (Appendix A). In total, 14 patients received SARS-CoV-2 specific treatment: 12 patients were treated by Monoclonal antibodies (MoAbs) and 2 patients received the PF-07321332 (Paxlovid) antiviral drug due to MoAbs unavailability. Only 1 patient refused to be treated for his infection. Timing of SARS-CoV-2 infection post-immunization is graphically reported in Figure 2B, showing a shorter, although not significant, time between immunization and post-immunization SARS-CoV-2 infection in the NR group.

T cells subsets. In comparison to HD, CVID patients had lower counts of lymphocytes (*p* < 0.0001) and CD4+ T cells (*p* < 0.0001). Moreover, CVID had lower CD4+ naive cells and central memory T cells (*p* < 0.0001). Of note, CVID patients displayed a lower CD4+ naïve/memory ratio in comparison to HD, indicating an imbalance in CD4 T cell subsets in CVIDs in favor of experienced T-cell subsets (*p* < 0.0001) (Figure 3A). 

Among CVID, the NR group had the lowest counts of lymphocytes, CD4+ T-cells, and CD4+ naive T cells (Figure 3B and Table 2). However, only four patients displayed a severe reduction in CD4+ (<200 cells/mm^3^) and/or CD4 naive cells (20 cells/mm^3^), thus fulfilling the criteria for the diagnosis of Late-Onset Combined Immunodeficiency (LOCID) [10]: 3/17 patients in the NR group, 1/18 in the IgG-R and none in IgG/igA-R group.

Patients carrying the pathogenic heterozygous variants of the PI3Ks gene and the TNFRSF13B gene were non-responders, as well as patients carrying the VUS(s), the heterozygous variant in the STX11 gene and the compound heterozygous of TNFRSF13C gene and the heterozygous VUS in the FAS gene. Patients with the heterozygous variant in the PLCG2 gene and the patient carrying VUS in the DCLRE1C, plus the IGLL1 genes, were classified as S1-IgG-R. Patients with the heterozygous VUS in the TNFRSF13C and TNFRSF13B genes, plus the heterozygous pathogenic variant of the LRBA gene, were classified as S1 IgG/IgA-R. 

Similar alterations were observed in the CD8 T-cells. CD8+ T cells and CD8 naive T cells were significantly reduced in CVID patients compared to HD (*p* = 0.023 and *p* < 0.0001, respectively) (Figure 4A), with an expansion of the antigen experienced T-cell subset, as shown by reduced naive/memory *ratio* (*p* < 0.0001) (Figure 4A). Among CVID, the NR group had the lowest counts of CD8+ naive T cells and the lowest CD8+ naive/memory *ratio* value (Figure 4B and Table 2).

*CD40 ligand expression.* Expression of the costimulatory molecule CD40L in CD4+CD45+RO+ T cells is known to be crucial for B-cell IgG production. Overall, after stimulation with CytoStim, CD40L expression was significantly reduced in CVID patients compared to HD (*p* < 0.0001) (Figure 3A). Similarly, CD40L expression on CD8+CD45+RO+ T cells was reduced in the CVID group (*p* = 0.0075) (Figure 3A).

Figure 3B and Figure 4B and Table 2 show the results in the CVID groups. The lowest values of CD40L expression were recorded in CD4+CD45+RO+ and in the CD8+CD45RO+ memory T cells of the NR group, indicating a severe impairment of T cell activation in the CVID patients unable to secrete anti-SARS-CoV-2 antibodies (Figure 3B and Figure 4B).

*Cytokine production.* After T-cell stimulation with CytoStim, CD4+CD45RO+ T-cells expressing TNFα were lower in CVID patients compared to HD, whereas the count of CD4+CD45RO+ T-cells expressing IFNγ was similar (Figure 3A,B). An increase in CD8+CD45RO+ expressing TNFα and IFNγ was observed in CVID after T-cell stimulation with CytoStim (IFNγ CD8+RO+: *p* = 0.0141; TNFα CD8+RO+: *p* = 0.0141). The highest number of cells expressing IFNγ and TNFα were observed in the CVID NR and IgG-R groups (Figure 4B, Table 2), indicating a more pronounced inflammatory signature in CVID with impaired antibody responses to SARS-CoV-2 vaccination.

*T-cell activation in response to Spike peptide stimulation*. Intracellular TNFα and IFNγ staining demonstrated specific CD4+ T-cell responses in 36% and 28% of immunized patients (*n*  =  17 and *n* = 13, respectively) after Spike peptide stimulation (median (IQR); TNFα: HD: 1.0 cell/mm^3^ (0.6–1.5), CVID: 0 cell/mm^3^ (0–0.5), *p* = 0.0002; IFNγ: HD: 0.75 (0.7–1.3), CVID: 0 cell/mm^3^ (0–0.3), *p* < 0.0001). All HD (*n*  =  7) had specific CD4+ T-cell responses to Spike peptide stimulation (Figure 5A). CVID groups developed similar counts of TNFα or IFNγ specific CD4+ T cells in response to Spike peptide stimulation (Figure 5B). Neither CVID patients nor HD showed measurable CD8+ T-cell responses (data not shown).

## 4. Discussion

CVID is the most frequent symptomatic primary immunodeficiency. In most CVID cases, the genetic causes remain undefined and the diagnosis is predominantly based on hypogammaglobulinemia with reduced MBCs and altered antibody production. CVID clinical phenotypes are broad, ranging from only bacterial infections to progression, to severe phenotypes similar to Combined Immunodeficiency [11]. The type and severity of the immune defects determine the efficacy of vaccines, with varying levels of impairment, ranging from normal to incomplete or absent responses [12,13]. Although not mandatory to diagnosis, vaccine challenge responses are an integral part of the diagnostic work-up of patients with suspected CVID [14] and might be used as a prognostic marker [15]. Antibody defects may also result primarily from impairment of the T cell compartment with a loss of T-cell function, low frequency of circulating CD4 T cells, naive CD4 T cells, and impaired activation and secretion of cytokines [6].

Data on the immunogenicity of the SARS-CoV-2 vaccine in patients with CVID have been available since the beginning of the pandemic [4,5,16]. Recently, seroconversion rates in patients with CVID have been shown to be lower than in healthy controls in a large prospective study on 196 CVID immunized adults, accounting for 50% [17]. The heterogeneity of the immune defects might account for the variable responses to COVID-19 vaccines [18].

Since vaccination became available, Italian CVID patients have been immunized mostly with mRNA COVID-19 vaccines [19]. We have previously shown a low rate of immune response after full immunization, as one third of CVID did not seroconvert and one third did not generate specific T cells, while the antibody response was boosted in convalescent patients by immunization [4,5], a figure similar to that described earlier by Hagin et al. in SARS-CoV.2 infected patients [20]. Differently, after the booster dose, a higher number of CVID responded by increasing the antibody levels and the frequency of specific memory B cells and T cells [21]. This heterogeneity of impaired immune responses underlines the need to link the different capacity to mount an antibody response to immunization to the underlying immunological abnormalities described in CVID [11].

We show here that CVID-associated T cell abnormalities may contribute to the inability to produce specific antibodies after SARS-CoV-2 full immunization. CVID non-responders showed a more complex disease phenotype characterized by bronchiectasis and autoimmune manifestations, including autoimmune cytopenia, and had a high rate of SARS-CoV-2 infection post-vaccination. The non-responder status was characterized by low average peripheral blood lymphocyte counts, low B cells, reduced MBCs, low naïve CD4+ and CD8+ T cells, and central memory T cells. In addition, CD4+ and CD8+ T cells were less able to express the CD40L, as summarized in Table 3.

While the CD40L is indispensable for the generation of the germinal center response after immunization, TNFα is required for the formation of primary follicles [22], being essential for the development and function of follicular dendritic cells [23]. Differently from HD, where the SARS-CoV-2 mRNA vaccine elicits a potent adaptive immune response in the absence of IFN-mediated inflammation [24], the CVID non-responders had a high frequency of CD8+IFNγ+ T-cells, indicating a T_H_1-driven immune dysregulation [7,16,25] but also a defective IFNγ release by Spike peptide-stimulated cells. Thus, the functional dysregulation of T-cells might account for impaired T-B cells interaction and the lack of specific antibody responses, confirming the role of germinal center reaction on protective antibody generation [26,27]. On the other hand, patients who produced both anti S1 IgG and IgA antibodies showed a less perturbed clinical and immunological phenotype, indicating a less impaired germinal reaction and a less pronounced inflammatory signature.

Since the beginning of the pandemic, T-cell responses have been found helpful in limiting virus replication and removing virus-infected cells [28]. Spike-specific T-cells responses were induced in CVID patients with a variable frequency [29], differently from T cells produced after multiple exposures to viral antigens by annual influenza virus immunization [30]. A possible explanation for this may be that since SARS-CoV-2 is a novel pathogen, the first antigenic stimulations by immunization might be not sufficient to induce a robust T-cell response in CVID. On the contrary, the majority of the immunocompetent subjects vaccinated individuals developed T cell responses, with variable results observed only in aged individuals [31].

This study has several limitations, mainly due to the low number of age- and sex-matched HD for the analysis of specific T cell responses. A second limitation is the lack of data on SARS-CoV-2 viral load, since we did not regularly measure the viral load of infected individuals, making the comparison between CVID patients and non-immunocompromised infected subjects difficult.

Two years later, COVID-19 is still a matter of concern for people with a compromised immune system. It is possible to hypothesize that when the pandemic turns into an endemic infection for the general population, in patients with CVID, as in the other fragile patients, the risk of infection might remain high. A proportion of them do not respond to COVID vaccines, so the search for a strategy for protection is necessary. Since infection is high in our subgroup of non-responders, at present, CVID patients might benefit from new strategies based on prophylaxis with SARS-CoV-2 monoclonal antibodies [32,33], and early antiviral drugs administration.

## Figures and Tables

**Figure 1 cells-11-01918-f001:**
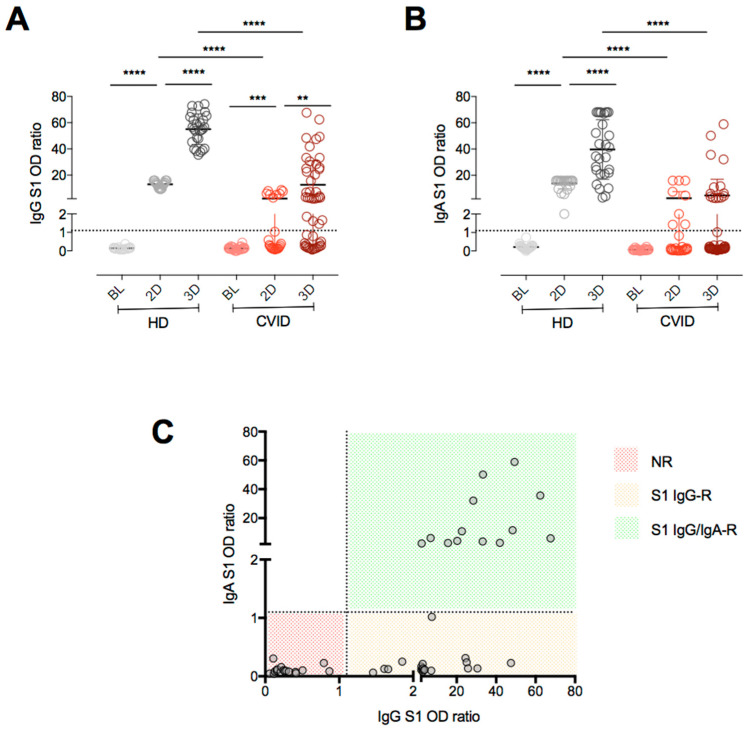
Anti-Spike S1 IgG (**A**) and IgA antibodies (**B**) in HD and CVID patients before and after full immunization with two doses (2D), and after the booster dose (3D) of mRNA BNT162b2 vaccine. Based on antibody responses after the booster dose, we identified three CVID groups (**C**): non responders (NR); anti-S1 IgG only responders (IgG-R), and anti-S1 IgG and anti-S1 IgA responders (IgG/IgA-R). Medians are plotted as horizontal bars and statistical significance were determined using two-tailed Mann–Whitney U-test or Wilcoxon matched-pairs signed-rank test. ** *p*  <  0.01, *** *p*  <  0.001; **** *p*  <  0.0001. HD *n* = 26; CVID *n* = 47.

**Figure 2 cells-11-01918-f002:**
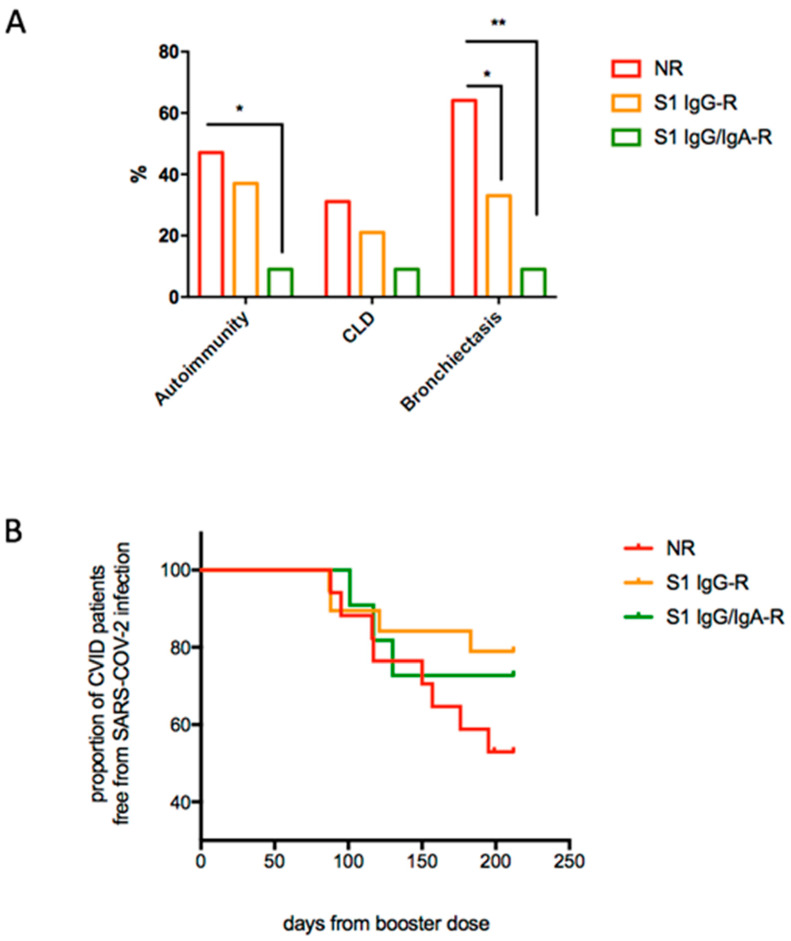
CVID-associated clinical manifestations (**A**) and proportion of CVID patients free from SARS-CoV-2 infection after the booster dose of the mRNA BNT162b2 vaccine (**B**). (**A**): bars represented frequencies of CVID-related complications. Statistical significance was determined using a two-tailed Chi-square test. (**B**): Time to COVID-infection after the third dose of vaccine administration was assessed by using Kaplan–Meier product-limit estimates and based on a log-rank and Gehan–Breslow–Wilcoxon test (difference among groups not significant). * *p*  <  0.05, ** *p*  <  0.01 (NR *n* = 17; IgG-R *n* = 18, IgG/IgA-R *n* = 12).

**Figure 3 cells-11-01918-f003:**
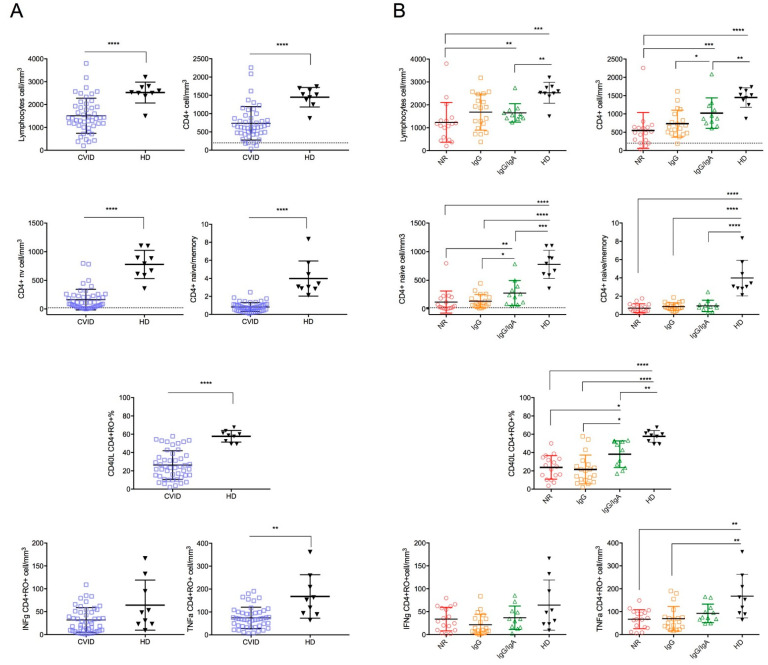
Peripheral lymphocytes count, CD4+ T cell subsets count, CD40L, TNFα and IFNγ expression in CD45+CD4+RO+ in CVID and HD. Comparisons between CVID and HD (**A**). Comparison between HD and CVID study groups (**B**). Levels of significance by Two-tailed Mann–Whitney U-test: **** *p* < 0.0001, *** *p* < 0.001, ** *p* < 0.01, * *p* < 0.05.

**Figure 4 cells-11-01918-f004:**
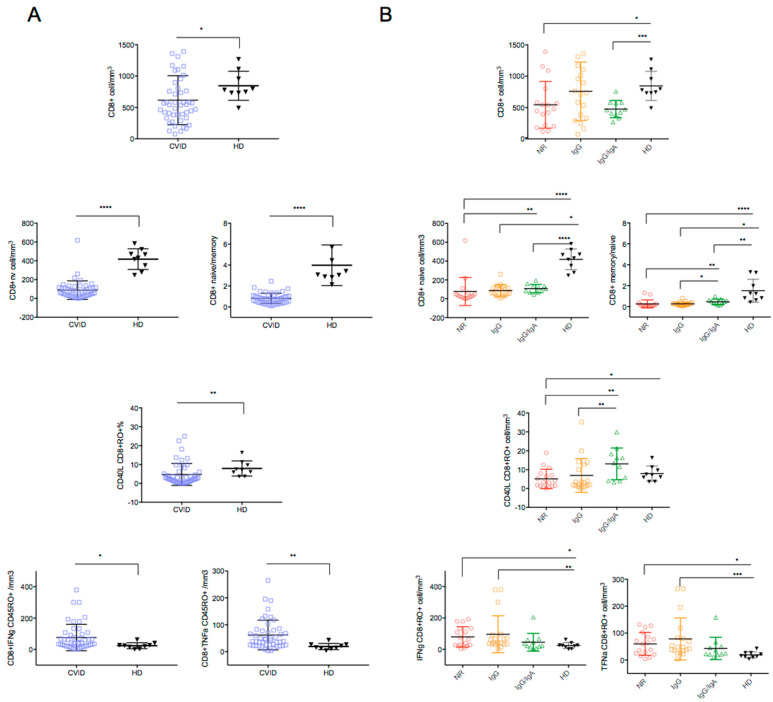
CD8+ T cell subsets count and TNFα and IFNγ expression in CD45+CD8+RO+ in CVID and HD (**A**) and comparison among HD and CVID groups is shown (**B**). Levels of significance by Two-tailed Mann–Whitney U-test: **** *p* < 0.0001, *** *p* < 0.001, ** *p* < 0.01, * *p* < 0.05.

**Figure 5 cells-11-01918-f005:**
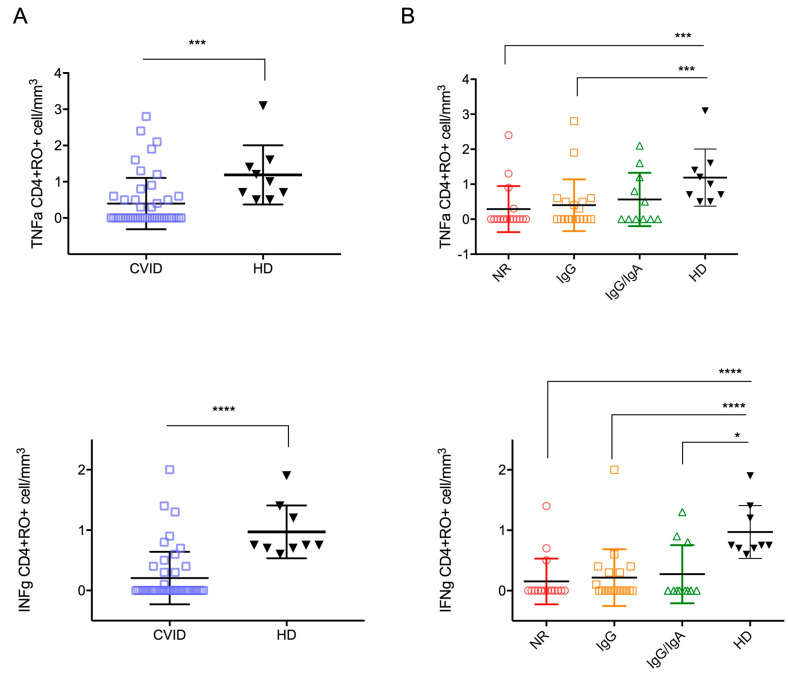
TNFα and IFNγ expression in CD4+CD45RO+ T-cells by intracellular flow cytometry after Spike peptide stimulation in CVID patients and HD (**A**), and in CVID patients grouped by their antibody response following third BNT162b2 mRNA COVID-19 vaccination (**B**). Levels of significance by Two-tailed Mann–Whitney U-test: **** *p* < 0.0001, *** *p* < 0.001, * *p* < 0.05.

**Table 1 cells-11-01918-t001:** Characteristics of CVID patients classified on the basis of anti S1 IgG- and IgA-antibody production following the third BNT162b2 mRNA COVID-19 vaccination dose.

							*p* Value		
	NR *n* = 17		S1 IgG-R *n* = 18		S1 IgG/IgA-R *n* = 12		NR vs. IgG	NR vs. IgG/IgA-R	IgG vs. IgG/IgA
Age (year), median (IQR)	56	(43–82)	46	(34–80)	59	(34–74)	0.166	0.562	0.166
Sex (female), *n* (%)	10	(59)	11	(61)	6	(50)	0.485	1.000	0.421
IgG (mg/dL), median (IQR)	670	(600–800)	600	(510–951)	780	(630–1130)	0.023	0.028	0.023
IgA (mg/dL), median (IQR)	2	(0–21)	3	(2–44)	29	(0–181)	0.071	0.068	0.071
IgM (mg/dL), median (IQR)	3	(0–142)	13	(2–75)	66	(1–117)	0.026	0.906	0.026
Lymphocytes (cells/mm^3^), median (IQR)	54.5	(20.5–74.1)	50.9	(27.9–73.8)	49.8	(1.53–70.7)	0.954	0.471	0.47
CD19+ (cells/mm^3^), median (IQR)	19.8	(1.63–83.66)	47.1	(11.9–161.5)	86.86	(22.5–208.9)	0.0083	<0.0001	0.134
MBC (cells/mm^3^), median (IQR)	1.52	(0–7.5)	6.27	(2.4–52.2)	37.93	(6.45–92.18)	0.0016	<0.0001	0.0245
IgM MBC (cells/mm^3^), median (IQR)	1.49	(0–7.5)	6.08	(2.3–44.7)	26.1	(3.89–73.56)	0.0021	<0.0001	0.0741
Switched MBC (cells/mm^3^), median (IQR)	0.01	(0.00–0.27)	0.15	(0.1–5.8)	5.36	(1.93–16.54)	0.0038	<0.0001	0.0004

**Table 2 cells-11-01918-t002:** Count of peripheral lymphocytes, CD3+, CD4+ and CD8+ subsets, CD40L in CD45+CD4+RO+ (expressed as percentage), and count of CD45+CD4+RO+ and CD45+CD8+RO+ expressing TNFα and IFNγ in CVID study groups. Values are reported as median (IQR).

	CVID (All) *n* = 47	NR *n* = 17	S1 IgG R *n* = 19	S1 IgG/IgA R *n* = 11	HD *n* = 9
Lymphocytes (cells/mm^3^)	1400 (1070–1930)	1170 (525–1375)	1870 (970–2470)	1550 (1420–1770)	2530 (2438–2775)
CD3+ (cells/mm^3^)	1112 (758–1617)	863 (451–1198)	1280 (738–1930)	1112 (966–1314)	1960 (1767–2119)
CD3+CD4+ (cells/mm^3^)	641 (470–876)	490 (243–636)	641 (470–998)	876 (749–1127)	1518 (1311–1664)
CD4+ naive (cells/mm^3^)	107 (33–229)	32 (14–194)	107 (53–201)	241 (87–395)	791 (600–997)
CD4+ central memory (cells/mm^3^)	146 (89–201)	109 (55–161)	140 (121–216)	173 (64–291)	376 (238–418)
CD4+ naïve/memory ratio	0.76 (0.41–1.05	0.56 (0.305–1.055)	0.78 (0.62–1.15)	0.83(0.55–1.05)	3.07 (2.855–5.07)
CD40L expression in CD45RO+CD4+ cells (%)	23.4 (14–36.7)	24.7 (12.8–33.95)	20.1 (9.9–26.9)	42.8 (23.4–51.8)	58.8 (50.9–62.4)
TNFα CD45RO+CD4+ (cells/mm^3^)	68.5 (37.2–98.4)	66.8 (30.5–98)	55.4 (31.3–77)	75.5 (66.2–117.9)	152.8 (91.7–233.5)
IFNγ CD45RO+CD4+ (cells/mm^3^)	27 (8.5–51.3)	29 (9.8–59.7)	10.1 (5.6–35.3)	34.3 (15.7–52.3)	48 (23–113.6)
CD3+CD8+ (cells/mm^3^)	542 (333–804)	482 (210.5–693.5)	736 (333–1106)	461 (382–579)	778 (730–1033)
CD8+ naive (cells/mm^3^)	64 (31–117)	28 (12–56.5)	85 (34–117)	106 (68–134)	431 (313.5–496)
CD8+ central memory (cells/mm^3^)	19 (10–33)	17 (9.5–30.5)	24 (13–38)	19 (10–36)	19 (11.5–27)
CD8+ naïve/memory *ratio*	0.76 (0.41–1.05)	0.11 (0.06–0.24)	0.23 (0.11–0.3)	0.47 (0.27–0.63)	1.23 (0.71–2.61)
CD40L expression in CD45RO+CD8+ cells (%)	2.6 (0.9–6.1)	3.4 (1.35–7.5)	3.0 (1.5–12.9)	13.5 (4.1–18.3)	7.2 (4.8–10.5)
TNFα CD45RO+CD8+ (cells/mm^3^)	42 (24.7–79.8)	59.7 (22.75–94.25)	48.1 (34.5–82.6)	24.8 (20.7–49.8)	15.1 (11.05–26.25)
IFNγ CD45RO+CD8+ (cells/mm^3^)	42.7 (22.8–97)	63.4 (23.95–133.0)	42.4 (33.4–91.1)	22.8 (12.2–59.7)	24.4 (8.85–33.7)

**Table 3 cells-11-01918-t003:** Summary of immune alterations and clinical conditions relevant to the lack of antibody response to SARS-CoV-2 immunization in CVID.

Peripheral blood Lymphocyte counts	Reduced
Circulating B cells	Reduced
Switched memory B cells	Reduced
Circulating CD4 T cells	Reduced
Circulating Naive CD4 and CD8 T cells	Reduced
CD40L expression in stimulated CD4+CD45+RO+	Reduced
TNFα and IFNγ expressing CD8+ cells	Increased
IFNγ release by SARS-CoV-2-induced CD4+ T-cells	Reduced
Chronic Lung disease, Bronchiectasis	Increased
Autoimmune cytopenias	Increased

## Data Availability

The data presented in this study are available on request from the corresponding author.

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
