# Peer review of "T-Cell Defects Associated to Lack of Spike-Specific Antibodies after BNT162b2 Full Immunization Followed by a Booster Dose in Patients with Common Variable Immune Deficiencies"

_cells, 2022, doi:10.3390/cells11121918_

Round 1

Reviewer 1 Report

Introduction: The authors need to discuss more the immune defects (including T cells) that occur in CVID patients. 

In their Figure 2, they need to define the conditions "inflammation and CPD" that are associated with low or non-responders. No statistical analysis for Figure 2B.

The rationale for genetic analysis (Results section) is not clear. The cohort is small and conclusions cannot be soundly drawned. The authors did not discuss this issue deeply.

Only 9 HD were analyzed with respect to T cell responses (T cell activation experiments; Figure 5). Are these experiments age- and sex-matched controlled? 

Did the authors studied the functional responses of CD8+ T cells as done for CD4+ T cells (Figure 5)? do they express less granzyme B and Fas-Ligand following in vitro activation and third boost vaccination? 

Not clear how increased TNF and IFNg CD8+ T cells in CVID patients after boost will contribute to non-responders? These cells should be protective against infection or severe disease?

It seems that all patients that were infected with SARS-CoV-2 subsequently to vaccination had asymptomatic/mild disease. Was it the Omicron variant? It is not clear whether the described immune defects contribute to severe disease or just for the risk of being infected. This should be more discussed.  

Author Response

Reviewer n.1

  • The authors need to discuss more the immune defects (including T cells) that occur in CVID patients. 

Response: In the Introduction, we summarized and referenced T cell defects in CVID as follow: “However, besides the impaired production of specific antibodies and memory B cells, abnormalities of T cell responses may also play an important role in vaccine failure. CD4 T-cell lymphopenia is associated with an increased risk of complications in CVID [6], as well as cytokine alterations, lack of IL-2, increased TNFα, IFNγ abnormalities with an IFN mRNA signature [7,8], mainly described in CVID patients with an inflammatory phenotype. We included in the discussion the reference n. 11 (11.Bonilla FA, Barlan I, Chapel H, Costa-Carvalho BT, Cunningham-Rundles C, de la Morena MT, Espinosa-Rosales FJ, Hammarström L, Nonoyama S, Quinti I, et al. International Consensus Document (ICON): Common Variable Immunodeficiency Disorders. J. Allergy Clin. Immunol. Pract. 2016, 4, 38-59) where there is a wide explanation on T and B cell defects in CVID.

  • In their Figure 2, they need to define the conditions "inflammation and CPD" that are associated with low or non-responders. No statistical analysis for Figure 2B.

Response. Thank you for this comment. We agree that a better definition of CVID-associated conditions is needed. We draw a new Fig. 2, including the statistical analysis. We specified: “Moreover, patients in the NR group in comparison to IgG/IgA antibody responders showed higher frequencies of autoimmune manifestations including cytopenias (p=0.049), and bronchiectasis (p=0.011) (Figure 2A), revealing a more complex clinical phenotype”. Moreover, we specified: “Timing of SARS-CoV-2 infection post-immunization is graphically reported in Figure 2B, showing a shorter, although not significant, time between immunization and post-immunization SARS-CoV-2 infection in the NR group”.

  • The rationale for genetic analysis (Results section) is not clear. The cohort is small and conclusions cannot be soundly drawned. The authors did not discuss this issue deeply.

Response. We followed the actual position of IUIS and we did genetic testing accordingly: “Genetic studies to investigate monogenic forms of CVID or for disease-modifying polymorphisms are not generally required for diagnosis and management in most of CVID patients, especially those who present with infections only without immune dysregulation, autoimmunity, malignancy, or other complications. In these latter groups of patients, however, single gene defects may be amenable to specific therapies (eg, stem cell therapy) and molecular genetic diagnosis should be considered when possible.” As specified in the text: In our cohort patients carrying the pathogenic heterozygous variants of the PI3Ks gene and the TNFRSF13B gene were non-responders as well as patients carrying the VUS(s), the heterozygous variant in the STX11 gene and compound heterozygous of TNFRSF13C gene and the heterozygous VUS in the FAS gene. Patients with the heterozygous variant in the PLCG2 gene and the patient carrying VUS in the DCLRE1C plus the IGLL1 genes were classified as S1-IgG-R. Patient with the heterozygous VUS in the TNFRSF13C and TNFRSF13B genes plus the heterozygous pathogenic variant of the LRBA gene was classified as S1 IgG/IgA-R. We think that this information, although not discussed further, might be useful for clinical immunologists for future comparison.

  • Only 9 HD were analyzed with respect to T cell responses (T cell activation experiments; Figure 5). Are these experiments age- and sex-matched controlled? 

Response. We included data on 9 HD immunized by three doses of vaccine, age- and sex- matched controls. We specified in the text and we added a sentence as a limitation of the study

  • Did the authors studied the functional responses of CD8+ T cells as done for CD4+ T cells (Figure 5)? do they express less granzyme B and Fas-Ligand following in vitro activation and third boost vaccination? 

Response.  We inserted in the results section, in the figure and in the Table 2,the functional responses of CD8+ cells, and in particular the expression of CD40L after stimulation in CD8+CD45RO+ cells in the different groups as done for CD4+ cells: “The lowest values of CD40L expression were recorded in CD4+CD45+RO+, and in CD8+CD45RO+ memory T cells of the NR group indicating a severe impairment of T cell activation in the CVID patients unable to secrete anti-SARS-CoV-2 antibodies (Figure 3B and 4B, and Table 2)”. We do not have data on granzyme B.

  • Not clear how increased TNF and IFNg CD8+ T cells in CVID patients after boost will contribute to non-responders? These cells should be protective against infection or severe disease?

Response.  We thank the Reviewer for this question. The dabate on the protective role of TNF and IFNg CD8+ T cells is still open also in HD. Most of the papers describe a protective or harmful role according to the timing of the COVID infection. As shown in Fig. 4, NR have a higher TNF and INFg production after stimulation than HD and CVID responders, possibly contributing to the non responders status.

  • It seems that all patients that were infected with SARS-CoV-2 subsequently to vaccination had asymptomatic/mild disease. Was it the Omicron variant? It is not clear whether the described immune defects contribute to severe disease or just for the risk of being infected. This should be more discussed.

Response. Thanks you for asking. Data on patients infected with SARS-CoV-2 subsequently to vaccination with asymptomatic/mild disease in our cohort (referreing to the pre-Omicron period) have been already published.  However, the vast majority of infections after four doses vaccination occurred after Omicron. This did not changed the clinical picture of a mild/asyptomatic infection in the majority of CVID patients.

Reviewer 2 Report

Covid-19 has emerged as a pandemic in 2020. It has affected majority of the population in many countries. Elderly people and people with immunodeficiencies are vulnerable population to covid-19 infection. Common variable immunodeficiency (CVID) is caused by a combination of genetic environmental factors, the patients have hypogammaglobulinemia. Patients with immune deficiencies are more susceptible to viral/bacterial infections. Patients with immune deficiencies have impaired antibody responses after immunization with vaccines.  A detailed analysis of CVID-associated T-cell defects contributing to the impaired antibody response to SARS-CoV-2 immunization in CVID is lacking. In this manuscript Pulvirenti et al have shown that defects in the production T- cell cytokines have a profound impact on SARS-CoV-2 specific antibodies after full immunization. Overall it is a very well designed study with very clear results.

Major Comments:

1.    In the figure 2A authors have shown how inflammation related factors change among the non-responders and S1 IgG/IgA responders. It is not clearly mentioned in the methods how these parameters were measured. Kindly provide those details in the methods section.

2.   Is viral load measured among the infected individuals ? Is it different between the CVID and non CVID patients? 

3.     Are there any differences in the antibody production between the Covid-19 infected CVID and uninfected CVID patients.

4.    Are the innate cytokines like IL-6, IL-12 are defective in CVID patients compared to HD? Because IL-6, IL-12 have a profound effect on B cell function. 

  Minor comments:

In page 5 "Cell isolation and cryopreservation, T and B cell phenotypes" method line 8 please remove thanks to the expression of igM . I am not sure what authors meant by this.

Author Response

Major Comments:

  • In the figure 2A authors have shown how inflammation related factors change among the non-responders and S1 IgG/IgA responders. It is not clearly mentioned in the methods how these parameters were measured. Kindly provide those details in the methods section.

Response. Thank you for this comment. We agree that a better definition of CVID-associated conditions is needed. We draw a new Fig. 2, including the statistical analysis. We specified: “Moreover, patients in the NR group in comparison to IgG/IgA antibody responders showed higher frequencies of autoimmune manifestations including cytopenias (p=0.049), and bronchiectasis (p=0.011) (Figure 2A), revealing a more complex clinical phenotype”. We inserted in the Methods a new paragraph on the statistical analysis.

  • Is viral load measured among the infected individuals? Is it different between the CVID and non CVID patients?

Response: We agree that this might be very intersting issue. However, we did not regularly measure the viral load of infected individuals, thus we cannot provide an exact figure. We added this comment as a study limitation.

  • Are there any differences in the antibody production between the Covid-19 infected CVID and uninfected CVID patients.

Response. We have already published data on differences in the antibody production between the Covid-19 infected CVID and uninfected CVID patients in the same special issue of Cells: B Cell Response Induced by SARS-CoV-2 Infection Is Boosted by the BNT162b2 Vaccine in Primary Antibody Deficiencies. Cells. 2021 Oct 27;10(11):2915. doi: 10.3390/cells10112915.

  • Are the innate cytokines like IL-6, IL-12 are defective in CVID patients compared to HD? Because IL-6, IL-12 have a profound effect on B cell function.

Response: We agree on the interest of this issue. However, we did not checked IL-6 and IL-12 production due to the limitation of the blood sample taken from patients and HD.

Minor comments:

  • In page 5 "Cell isolation and cryopreservation, T and B cell phenotypes" method line 8 please remove thanks to the expression of igM . I am not sure what authors meant by this.

Response: We specified that the expression on IgM on CD27+ memory B cells allows to identified the IgM memory B cell substet. 

Round 2

Reviewer 1 Report

addressed